# Probiotics and Prebiotics for the Amelioration of Type 1 Diabetes: Present and Future Perspectives

**DOI:** 10.3390/microorganisms7030067

**Published:** 2019-03-02

**Authors:** Sidharth Mishra, Shaohua Wang, Ravinder Nagpal, Brandi Miller, Ria Singh, Subhash Taraphder, Hariom Yadav

**Affiliations:** 1Department of Internal Medicine-Molecular Medicine, Centre for Diabetes, Obesity and Metabolism, Wake Forest School of Medicine, Winston-Salem, NC 27101, USA; spmishra@wakehealth.edu (S.M.); shaowang@wakehealth.edu (S.W.); rnagpal@wakehealth.edu (R.N.); bcmiller@wakehealth.edu (B.M.); ria.singh@gmail.com (R.S.); 2Department of Microbiology and Immunology, Centre for Diabetes, Obesity and Metabolism, Wake Forest School of Medicine, Winston-Salem, NC 27101, USA; 3Department of Animal Genetics and Breeding, West Bengal University of Animal and Fishery Science, Kolkata 700 037, India; subhash.taraphder@gmail.com

**Keywords:** autoimmune, diet, diabetes, prebiotics, probiotics, fiber, gut, microbiota, microflora

## Abstract

Type 1-diabetes (T1D) is an autoimmune disease characterized by immune-mediated destruction of pancreatic beta (β)-cells. Genetic and environmental interactions play an important role in immune system malfunction by priming an aggressive adaptive immune response against β-cells. The microbes inhabiting the human intestine closely interact with the enteric mucosal immune system. Gut microbiota colonization and immune system maturation occur in parallel during early years of life; hence, perturbations in the gut microbiota can impair the functions of immune cells and vice-versa. Abnormal gut microbiota perturbations (dysbiosis) are often detected in T1D subjects, particularly those diagnosed as multiple-autoantibody-positive as a result of an aggressive and adverse immunoresponse. The pathogenesis of T1D involves activation of self-reactive T-cells, resulting in the destruction of β-cells by CD8^+^ T-lymphocytes. It is also becoming clear that gut microbes interact closely with T-cells. The amelioration of gut dysbiosis using specific probiotics and prebiotics has been found to be associated with decline in the autoimmune response (with diminished inflammation) and gut integrity (through increased expression of tight-junction proteins in the intestinal epithelium). This review discusses the potential interactions between gut microbiota and immune mechanisms that are involved in the progression of T1D and contemplates the potential effects and prospects of gut microbiota modulators, including probiotic and prebiotic interventions, in the amelioration of T1D pathology, in both human and animal models.

## 1. Introduction

Type 1-diabetes (T1D) is a chronic autoimmune disease diagnosed commonly during early childhood and is characterized by immune-mediated destruction of the insulin-producing pancreatic beta (β)-cells [1]. The prevalence of T1D is increasing worldwide, mainly due to the lack of successful preventive and therapeutic strategies. Hence, a more inclusive understanding of the pathophysiology and risk factors of T1D is needed to discover novel and successful preventive and/or therapeutic strategies. According to the EuroDIAB (European Diabetes) study, the incidence of T1D among children (boys and girls) of 0–4 years increased by 3.7%, 5–9 years increased by 3.4% in boys and 3.7% in girls and 10–14 years age groups increased by 3.3% in boys and 2.6% in girls per annum, respectively, over the last 25 years [2]. In addition, as per the combined data, the lifetime risk for occurrence of T1D has exceeded more than 1% in European and North American people [3]. The latest Environmental Determinants of Diabetes in the Young (TEDDY) study has reported that more than half of T1D cases are diagnosed after 20 years of age [3].

Genetic predisposition coupled with environmental factors are important elements in the development of T1D [3,4]. The early pathogenesis of T1D is characterized by an increased production of autoantibodies against β-cell antigens and also by insulitis (the increased inflammation in pancreas), which is followed by reduction in insulin secretion and β-cell death [4]. The precise environmental triggers that induce T1D pathogenesis remain unknown; however, some of the most commonly responsible factors are genetics and/or the environment [4,5]. Some environmental factors of T1D are viral infections, antibiotics, consumption of cereal or cow-milk proteins at a young age, decreased intake or endogenous synthesis of vitamin D, lack of breast-feeding, seasonality, exposure to endocrine-disrupting chemicals and other dietary factors [5,6,7] (Figure 1). In addition, the contribution of the microbial community living in the human and mouse gastrointestinal tract (gut microbiota) in T1D pathology has recently been elucidated. It has also been demonstrated that most of the above-described environmental factors related to T1D pathology can also influence gut microbiota or vice-versa [8]. Further, gut microbiota and its metabolites can influence the function of gut mucosal immune cells or instigate abnormal immune cell functioning such as autoimmunity against β-cells [8]. These inter-related physiological functions clearly point toward gut microbiota as an environmental factor contributing in T1D pathology; however, the precise mechanisms underlying this interaction remain unclear.

The gut microbiota composition and functions are highly diverse and complex and is predominated by bacteria belonging to the two major phyla *Bacteroidetes* and *Firmicutes*, particularly under normal and healthy milieus [9]. In both T1D human subjects and mouse T1D models, decreased *Firmicutes* and increased *Bacteroidetes* gut populations are commonly found, indicating a link between gut microbiota and T1D [10]. Abnormally increased ratio of *Bacteroidetes*:*Firmicutes* has been found to be associated with the development of T1D; although, contradicting results indicate that a decreased *Bacteroidetes*:*Firmicutes* ratio is associated with an obese phenotype, rather than a lean phenotype [11]. Thus, the influence of the *Bacteroidetes*:*Firmicutes* ratio remains controversial with regard to the development of diabetes. Differences in gut microbiota may appear due to varying glucose levels in host body fluids, which may result from the gastrointestinal environment or diet. Moreover, gut microbiota composition and function is impacted not only by the host diet, lifestyle and genetics but also by the mode of birth, i.e., vaginal delivery or Cesarean section (C-section) [5,6]. For example, the gut of babies that are delivered vaginally is predominated by bacteria seeded from maternal vaginal and perianal microbiota, including *Lactobacillus*, *Prevotella* and *Sneathia*. On the other hand, babies born via C-section are colonized heavily with bacteria belonging to the genera *Staphylococcus*, *Corynebacterium* and *Propionibacterium*, which are derived primarily from the maternal skin or the hospital environment and medical equipment [12,13]. During the first six months of life, infants born by C-section are also found to be colonized with several harmful microbes and opportunistic pathogens such as *Clostridium difficile, Clostridium perfringens and Clostridium* cluster I, while having abnormally lower population of several commensal bacteria including *Bacteroides fragilis*, *Bifidobacteria* and *Escherichia coli* [12,14,15]. Although, the precise mechanism(s) are not known, but these changes might be associated with the development of T1D, as decreased Bifidobacteria (common probiotics) can influence gut permeability and mucosal immune response influencing autoimmune reactions.

Gut microbiota interacts with the host cells via cellular components, such as lipopolysaccharides (LPS), metabolites including short-chain fatty acids (SCFAs; i.e., acetate, propionate and butyrate) and/or bile acids, as well as several other newly discovered metabolites [16]. Although, several mechanism(s) have been proposed to explain the interactions of host-microbiota interactions, the two majorly studied pathways are: (i) Toll-like receptors (TLRs) [17] and/or the nucleotide-binding oligomerization domain-like receptors (NLRs) [18], and (ii) the free-fatty acid receptors 2/3 (FFAR2/3) [19,20]. These signaling pathways could also be modulated by gut microbiota modulators, including probiotics and prebiotics [21]. By definition, probiotics are live bacteria that, when consumed in sufficient numbers, provide specific health benefits to the host [22,23]. Prebiotics are substrates that are selectively utilized by host microorganisms conferring health benefits [21,24]. Several previous reports have discussed the potential of probiotics and prebiotics in the context of T1D [10,24,25]; however, these reports are focused largely on either probiotics, prebiotics or gut microbiota and refer mainly to data from either animal models and/or preclinical or clinical studies. Therefore, comprehensive reports compiling data from both animal models as well as clinical studies and discussing the available literature related to the role of probiotics, prebiotics and gut microbiota in the pathophysiology, prevention and/or amelioration of T1D are lacking. In this context, the present manuscript aims to review and discuss detailed and updated information on how gut microbiota can influence the pathogenesis of T1D, while also discussing the present status and future prospects pertaining to the use of gut microbiota modulators for the amelioration of T1D progression. To our knowledge, this is the first report to review the role of both probiotics and prebiotics in the amelioration of T1D based on both human and animal studies and provide updated information on gut microbiota-immune axis in context of T1D, thereby bringing together important information and knowledge regarding the development of T1D as well as the therapeutic strategies to prevent and cure it.

All information related to T1D was collected from different literature databases including PubMed, Google Scholar, Science Direct, and general web search engines like Google and Yahoo. The keywords used for collecting this information were: Type 1-Diabetes, T1D and gut microbiota, T1D and gut microbiome, prebiotics and T1D, probiotics and T1D, human T1D, T1D mouse model, T1D rat model, T1D NOD mouse/mice, T1D BBRD mouse model, T1D and SCFAs, gut microbiota composition in T1D patients, gut microbe-immune interaction in T1D, immune-responses in T1D, pathogenesis of T1D, Peyer’s patches and GALT function in T1D, pancreas immunoresponse in T1D, T1D in children, T1D in humans, T1D in elderly, T1D and antibiotics, T1D and human leukocyte antigen, T1D and bovine leukocyte antigen, Lactobacillus and T1D, Bifidobacteria and T1D, microbiome and autoimmune, and inflammation, microbiota and T1D. Emphasis was placed on findings in the latest literature, published between 2015 and 2018. Wherever possible, we have directed the readers to old reviews on topics that are not directly relevant to be expanded in this manuscript.

## 2. Role of the Gut Microbiota-Immune Axis in T1D

The interaction between gut microbiota and host immune cells plays a critical role in the development of T1D [21,26]. Immune cells can sense the metabolites and antigens produced by gut microbes that can modulate the functions of immune cells and can either protect or accelerate the progression of T1D pathogenesis [24,26]. Similar to the gut, the pancreas also contains its own microbiota and the modulation of this ‘pancreatic’ microbiota is associated with the intra-pancreatic immune responses and induction of disease conditions including pancreatic cancer and T1D [27]. In this context, the subsequent section summarizes how gut microbiota and immune cells interact with each other and how unsolicited perturbations (dysbiosis) in these interactions can contribute to or instigate the pathogenesis of T1D (Figure 2).

### 2.1. The Pathogenesis of T1D

In T1D, pancreatic β-cells are destructed by immune attacks that are mediated majorly by cytotoxic T-cells; however, the mechanisms underlying how cytotoxic T-cells get activated are not fully described [26]. However, it is known that the developed islet of autoreactive T-cells becomes highly competent in destroying the healthy β-cells [28]. These episodes are followed by an increased infiltration of other immune cells (e.g., macrophages) thereby causing the development of insulitis (inflammatory islets) and ultimately insulin deficiency caused by increased β-cell death and declined β-cell mass [29]. The autoreactive immune response takes place in the pancreatic lymph nodes (PLNs), which constantly supply autoreactive T-cells and sense the β-cell antigens [29]. Dendritic cells (DCs) and macrophages act as innate immune effector cells and activate autoreactive cytotoxic T-cells [26], in addition to also serving as professional antigen-presenting cells (APCs) in the context of the Major Histocompatibility Complex class II (MHC II) molecules to induce T-cell activation, which contributes to T1D pathogenesis through an adaptive immune response [28]. This connection between the innate and adaptive immune systems indicates that the development of autoimmune T1D in the presence of genetic predisposition could be promoted by environmental factors [29]. Further, this suggests that innate immunity may be responsible for promoting the hostile adaptive immune response.

Once the T-cells become autoreactive in PLNs, the CD4+ T-cells continually proliferate and differentiate into auto-reactive CD4+ effector T-cells (Teffs) and further accelerate antigen recognition by APCs [30]. During these T-cell/APC interactions, Teffs develop and are further activated by the complement system [30,31]. These activated Teffs produce cytokines such as interferon (IFN)-γ and interleukin (IL)-2 in the pancreatic islets, which activate cytotoxic CD8^+^ T-cells and attract macrophages. The activation of cytotoxic CD8^+^ T-cells and the concomitantly increased recruitment of macrophages leads to insulitis [31], wherein these cytotoxic T-cells and macrophages further aggravate the production of cell death-inducing cytokines (and their receptor signaling), thereby inducing reactive oxygen/ nitrogen species, mitochondrial stress, and DNA damage that eventually results in β-cell death [28,30]. In addition, CD8+ T-cells also release granzymes or perforin, which are proteins that mediate direct toxicity to the β-cells [31]. Furthermore, B7-H4, a member of the B7-CD28 family of co-signaling molecules that is negatively correlated with T-cell activation, is also expressed less in the pancreatic islet of T1D in humans [32]. Expression of B7-H4 is closely associated with insulin secretion from islet β-cells [32]. Macrophages releasing pro-inflammatory cytokines (i.e., IL-1β, IFN-γ, tumor necrosis factor [TNF]-α) induce cytotoxic effects on the β-cells, thereby leading to the development of T1D [26,31]. However, it remains elusive as to where exactly does this autoimmune stimulation start.

Genetic predisposition is a major component that contributes in the progression of T1D; however, this predisposition must be triggered to induce autoimmune reactions [5,33]. One postulation is that these stimulations can start from the gut. The intestine and its microbiota remain in direct contact with the external environment and present a complex site for interacting with the mucosal immune system and, plausibly, also with gut-associated lymphatic tissues (GALT), which contain a large population of various kinds of immune cells [34]. GALT closely resemble other secondary lymphoid tissues and consist of Peyer’s Patches (PPs), the appendix, and the mesenteric lymph nodes (MLNs), and are distributed throughout the intestinal wall with multiple permanent or transient lymphoid follicles [35]. The PPs are oval-shaped or round lymphoid follicles located in the sub-mucosal layer of the ileum and extend to the mucosal layer. PPs are found at the lowest part of the human small intestine, mostly in the distal jejunum and ileum, with a minimal amount in the duodenum [36]. In humans, the number of PPs is highest during early adulthood, particularly between 15 and 25 years of age, and then starts declining with age [36]. The PPs consist of both APCs (e.g., DCs) and mononuclear cells (e.g., macrophages, T-cells and B-cells). The B-cells are found in the follicular germinal centers whereas T-cells are distributed in the zone between the follicles [37]. All the lymphoid follicles are covered with follicle-associated epithelium (FAE) [37], which is characterized by the presence of fewer mucus-producing goblet cells and Microfold cells (M-cells) [37]. The M-cells are associated with the uptake and transport of antigens from the lumen to the APCs. The dendrites of DCs are distributed through the transcellular M-cell specific pores [36,38]. Simultaneously, the paracellular pathway is tightly regulated to prevent the transmission of antigen to interact with the immune cells [38]. Moreover, in FAE, more tight junction proteins are expressed, thus rendering less permeability even to ions and macromolecules [38].

PPs play a significant role in the immune response within the intestinal mucosa by tracking the immune activity of the intestinal lumen in the presence of pathogenic microorganisms [37]. The MLNs connect the intestine to the pancreas through PLNs [35,39]. Specifically, the PPs of the intestine connect to the MLNs through apparent lymphoid circulation. The superior MLNs are ventrally connected to the inferior PLNs. The interaction of immune cells in PPs/GALT is associated with prominent changes appearing in the PLNs [40]. It remains undefined how the immune cells migrate from MLNs to PLNs and contribute to β-cell death; however, several reports have suggested that GALT immune cells are first encountered with certain environmental changes occurring in the gut [39]. For example, abnormal changes in gut microbiota and/or the overgrowth or encounter of pathogenic microorganisms in the intestine first activate macrophages, T-cells, B-cells or dendritic cells of PPs and other sites of GALT to develop an immune response [40]. These stimulated immune cells then migrate from GALT to PLNs and perform their corresponding functions therein. Normally, the B-cells and T-cells mature in the bone marrow and thymus, respectively. Further, the maturation of undifferentiated B-cells and T-cells that escape from the bone marrow and thymus takes place in the PPs. During this maturation and differentiation of immune cells and their exposure to the intestinal environment, the functions of local immune cells are significantly modulated [41]. For example, exposure of foreign antigen leads to increased secretion of immunoglobulin (Ig) A and cytotoxic CD4^+^ T-cells from the lamina propria [34]. The intestinal DCs localized in the PPs are responsible for the activation of IgA+ specific B-cells [41]. In T1D children, the integrity of the intestinal epithelium, including FAE is decreased, thereby leading to “leaky” gut, which allows antigens to easily penetrate the FAE and stimulate immune cells [39]. In response to immune-regulation, the active plasma B-cells, T-cells and DCs are re-circulated from PPs to MLNs through apparent lymphoid tissues [39]. At MLNs, the immune response is further amplified and eventually participates in the development of insulitis [39].

### 2.2. Gut Microbiota-Immune Interactions in T1D

Recent studies demonstrate that environmental factors, such as gut microbiota, interact closely with the immune system and contribute to the development of T1D [21,42]. It is well known that the interactions between gut microbiota and host immune cells contribute to the normal development and maturation of the immune system and its regulation [34,42]. Therefore, it is unsurprising that dysbiosis of gut microbiota and/or its metabolic activities could induce abnormal immune responses in the GALT, such as abnormally more secretion of IgA and proliferation of colonic regulatory T-cells (Tregs) [40]. The microbiota-induced impairment of the immune response in GALT can also impact the systemic immune response [40]. As mentioned above, it is postulated that gut microbiota regulate the host immune response through two well-known mechanisms: 1) by activating the innate immune response via TLRs [17] and/or 2) by activating FFAR 2/3 via microbial metabolites such as SCFAs (acetate, propionate and butyrate) and lactic acid [43,44,45]. Among these SCFAs, butyrate is known to be associated with the differentiation of naive T-cells into Tregs, while acetate and propionate are known to be essential for the migration of Tregs to the intestine [45]. Over-activation of TLRs and an abnormally low production of SCFAs, mainly butyrate, are known to have responsive effects on T1D-associated autoimmunity and could provide important therapeutic targets for the prevention of T1D [46].

TLRs are essential for the recognition of microbial molecules including nucleic acids, proteins and LPS that are derived from bacteria, fungi, viruses or other microbes [17]. In addition, TLRs can also identify the endogenous molecules released from the damaged tissues or cells through damage-associated molecular patterns (DAMPs) [45,47]. Ten TLRs (TLR1-10) in humans and 12 TLRs (TLR1-TLR9 and TLR11-TLR13) in mice have been identified. Most of these TLRs are present on the cell surface, except TLR3, 7, 8 and 9, which are expressed on the endosomal intracellular compartment. The activation of TLRs plays a significant role in the progression of T1D, either by sensing abnormal gut microbiota signals and/or sensing signals from the damaged pancreatic β-cells via DAMPs. During the progression of T1D, the innate immune system is activated via TLR2 and TLR4 [48]. However, some reports suggest that the TLRs (e.g., TLR3, 7 and 9) are expressed in the pancreas of the T1D patients [49]. In non-obese diabetic (NOD) mice with individual TLR knockouts, TLR2, 3 and 4 have been found dispensable in the pathogenesis of T1D [17,44]. However, in Bio-Breeding diabetes-resistant (BBDR) rats, T1D is developed by Kilham rat virus (KRV) through the activation of TLR2, 3, 4, 7, 8 and 9 [50]. All of these TLRs, except TLR3, are activated through the canonical myeloid differentiation primary response protein 88 (MyD88) pathway, which induces the production of inflammatory cytokines [51]. TLR pathways modulate the transcription nuclear factor kappa-light-chain-enhancer of activated B-cells (NF-κB) and the I kappa B kinase (IKK) complex [17]. NF-κB also acts as a regulator of inflammatory mediators such as IL-1β that are common determinants of T1D pathology [17].

On the other hand, in T1D, SCFAs-specific cell surface receptors viz. FFAR2 and FFAR3 are found to be expressed in various immune cells [52], plausibly because the number of SCFAs-producing microbes in the gut is dominated [52]. These SCFAs interact with the intestinal epithelial cells and are associated with an altered intestinal gene expression and its normal maturation by post-translational histone modifications [53]. FFAR2 and FFAR3 are G-protein coupled receptors (GPCRs) expressed on intestinal epithelial cells [19]. Specifically, FFAR2 is coupled with Gα_i_ and Gα_q_ while FFAR3 transmits the signal through the Gα_i_ [19,54]. Both FFAR2 and FFAR3 play a vital role in maintaining a healthy gut environment by regulating intestinal immune homeostasis [19,52]. FFAR2 and FFAR3 express a wide range of effects on intestinal immune homeostasis by regulating the intestinal immune barrier [20]. Activation of FFAR2 and FFAR3 by SCFAs influences the effect of inflammatory mediators on the intestinal epithelial cells [20,52]. This receptor signaling modulates the immune response by influencing the proliferation and differentiation of Tregs, which leads to changes in the array and magnitude of inflammation [55]. Along with this, infiltration of activated FFAR2 into the immune cells causes cytotoxic immune cell apoptosis. In addition, in T1D patients, FFAR2 remains upregulated in peripheral blood mononuclear cells [43], wherein it induces apoptosis of infiltrated pancreatic macrophages and maintains blood glucose homeostasis [43]. Subsequently, the immune response is transmitted from intestine to the pancreas via superior MLNs and inferior PLNs. In the pancreatic islet β-cells, FFAR2 and FFAR3 activation triggers the production of cathelicidin-related antimicrobial peptides (CRAMPs), which are responsible for immunomodulatory effects [56]. Moreover, FFAR2 is upregulated in the peripheral blood monocytes while FFAR3 is correlated with metabolic markers and inflammation [43,57]. The role of gut microbiota dysbiosis in T1D pathology and its modulation to restore gut homeostasis could prove to be beneficial in the amelioration of T1D. Alteration in gut bacterial metabolic products, such as 4-hydroxyhippuric acid and N-succinyl and L-diaminopimelic acids, affects intestinal permeability [58]. Metabolome analyses have reported that the kynurenine pathway gets over-activated in T1D children and this is associated with an increased production of tryptophan and phenylalanine derivatives [58].

Interaction of bacterial components, such as LPS or other metabolites, with adaptive and innate immune systems could provide a preventative measure against T1D [59,60]. In this context, the subsequent sections focus on the effects of probiotics and prebiotics on the gut microbiota-immune axis and their influence on the development and management of T1D. Notably, the immune system and pathology underlying T1D in humans is different from that in animal models, and these differences might also reflect the host-specific effects of probiotics and prebiotics interventions on the T1D phenotype. The elucidation of differences in the mechanisms of action of T1D in humans and animals will facilitate experimental research and aid in discovering novel therapeutic strategies to ameliorate the T1D. Some of the most common animal models used to study the effect of probiotics, prebiotics and drugs on T1D are NOD mice [61], Streptozotocin (STZ)-induced T1D rats and mice [62,63], Alloxan-induced T1D Swiss Webster mice [64], BBDR rats [65,66], and Bio-Breeding Diabetic Pathogen (BBDP) rats [65,66]. The interactions of probiotics and prebiotics with gut microbiota and immune system and their involvement in T1D pathology are discussed separately for human and animal models.

## 3. Probiotic Interventions to Ameliorate T1D

Probiotics are live microorganisms that, when administered in adequate amounts, provide a health benefit to the host [22,67]. Probiotics are an integral part of the human gut microbiota and help in maintaining a healthy gut microbiota homeostasis and the normal regulation of microbial metabolic activities, such as the production of beneficial SCFAs [24,68]. The consumption of specific probiotic strains results in several health benefits, including the normal regulation of gut membrane integrity and permeability, thereby preventing gut leakiness, endotoxemia, and inflammation [67]. Several probiotic strains are also known to regulate pro-inflammatory signaling pathways by suppressing TLR signaling [17,69]. Specifically, the consumption of selected probiotic strains has been found to decrease the level of pro-inflammatory cytokines including IL-6, IL-1β and TNF-α while increasing that of anti-inflammatory cytokines, such as transforming growth factor-β (TGF-β) and IL-10 [24,69]. Therefore, probiotics might also be helpful in preventing T1D. Some probiotic strains have also been found to confer beneficial effects on the host by increasing the production of beneficial metabolites via modulation of gut microbiota [68]. Also, the administration of specific probiotic strains could increase the production of SCFAs (e.g., butyrate) and thus might balance the intestinal cellular homeostasis by activating FFAR2 and FFAR3, which contribute in immune system regulation and the pathogenesis of autoimmune diseases such as T1D [20]. In addition, SCFAs-mediated activation of FFAR2/3 could also enhance the production of glucagon-like peptide-1 (GLP-1) from intestinal L-cells. GLP-1 is a hormone that stimulates the secretion of insulin from the pancreatic β-cells, thereby decreasing blood sugar levels (known as the ‘incretin effect’) [19,70,71]. These facts demonstrate the potential of probiotics in preventing/managing T1D via maintaining/restoring homeostasis of the gut microbiota-immune axis (Figure 3).

### 3.1. Animal Studies

In a study on C57BL/6 mice, the feeding of specific probiotic *Lactobacillus (L.) brevis* strains (*L. brevis* KLDS 1.0727 and *L. brevis* KLDS 1.0373) has been found to protect against STZ-induced T1D and reduce blood glucose levels via gamma-aminobutyric acid (GABA) [62]. In another study on STZ-induced diabetic mice, strains belonging to *Bifidobacterium (B.)* spp. were found negatively associated with β-cell autoimmunity [72], wherein the feeding of probiotics was associated with an elevated innate response through proteins, including protein kinase B (Akt), IκB kinase alpha (IKKα), nuclear factor-kappa B inhibitor alpha (IκBα) and extracellular-signal-regulated kinase 2 (ERK2) [72]. Akt may activate IKKα with stimulation of the IκBα factor, which reduces the response of NF-κB and consequently inhibits the transcription of pro-inflammatory cytokines [72]. The treatment of NOD mice with probiotic strains belonging to families *Bifidobacteriaceae* and *Lactobacillaceae* and genus *Streptococcus thermophilus* has been shown to ameliorate T1D via positive modulation of gut microbiota composition and reduced intestinal inflammation by maintaining gut immune homeostasis and inhibiting IL-1β expression [73]. In addition, the pro-tolerogenic components of inflammasome, such as indoleamine 2,3-dioxygenase (IDO) and IL-33, were also found to be simultaneously increased, while helping to maintain cellular balance in the gut mucosa, MLNs and PLNs through regulation of DC function and the Teff/Treg ratio [73]. In another study on STZ-induced diabetic mice, *L. reuteri* treatment protected bone loss and prevented the upregulation of TNF-α, thereby also suppressing the osteoblast Wnt10b pathway [74]. The feeding of a *L. lactis* strain has also been shown to have preventive effects against T1D progression in NOD mice via stimulation of secretion of anti-inflammatory cytokines, including IL-10 and (pre-) proinsulin [75,76]. Interestingly, the combination of *L. lactis* strain with low doses of anti-CD3 enhanced the production of IL-10, thereby preserving the function of β-cells. In addition, the treatment also led to the development of antigen-specific Foxp3^+^ Tregs, which maintains pancreatic islets and ameliorates T1D [75,76,77]. Also, in another study on STZ-induced C57BL/6 mice, the administration of probiotic strains *L. kefiranofaciens* M and *L. kefiri* K was shown to stimulate the production of IL-10 in the pancreas [78]. IL-10 production helps in restraining the levels of T-helper (TH) cell 1-associated cytokines (IL-1β, IL-6 and IL-2) and pro-inflammatory cytokines (TNF-α) present in the pancreas. Moreover, bacterial strains M and K were also responsible for an increase in the production of GLP-1 and the regulation of insulin synthesis from pancreatic islet β-cells [78]. In an NOD mouse model, the probiotic strains belonging to *Streptococcus salivarius* subsp. *thermophilus*, *Lactobacillus* spp. and *Bifidobacterium* spp. were shown to activate the cytokine secretion pattern from a pro-inflammatory state to an anti-inflammatory state in GALT [79]. This decreased the chances of occurrence of an islet-specific autoimmune condition by reducing insulitis and regulating the maintenance of the diversity of B-cells, thereby providing protection against autoimmune T1D [79]. Hence, the application of such probiotic strains could be envisaged to be helpful in the restoration of intestinal cells and insulin-secreting cells by maintaining a healthy gut microbiota spectrum, thereby aiding in the amelioration of hyperglycemia and autoimmune T1D.

Unlike BBDP rat models, BBDR rats show resistance to T1D [65,66]. When BBDR rats are exposed to KRV, they tend to develop T1D [65]. Similar findings have also been seen in LEW.1WR1 rats subjected to viral infections; they develop autoimmune T1D because it is associated with β-cell infection and upregulation of intra-islet innate immune cells [80,81]. Experimentally, it has been established that when BBDP rats are orally fed the *L. johnsonii* strain N6.2 (LjN6.2) isolated from BBDR rats, they become resistant to the occurrence of T1D [65]. However, when *L. reuteri* strain (LrTD1) is administered in a similar manner, BBDP rats develop T1D [65]. This disparity in T1D development arises primarily due to the biasness in the action of TH17 in LjN6.2-fed BBDP rats [65], which is associated with increased levels of cytokines such as IL-6 and IL-23 in the MLNs of LjN6.2-fed BBDP rats compared to LrTD1-fed BBDP rats [65]. Therefore, *L. johnsonii* could reduce the incidence of T1D in BBDP rat model and was associated with TH17 lymphatic cell predisposition within the MLNs. It also decreased the risk of T1D occurrence by increasing a high concentration of intestinal tight-junction protein claudin; *L. reuteri* did not release any of this protein [82]. In another study on STZ-induced diabetic rats, feeding probiotic-fermented milk consisting of *L. rhamnous* MTCC5957, *L. rhamnous* MTCC5897 and *L. fermentum* MTCC5898 has been shown to improve the health of these rats [83]. This was associated with decreased TNF-α and IL-6 levels but with no significant difference in the amount of TGF-β between control and treated rats [83]. The consumption of probiotic-fermented milk also helped in decreasing blood glucose levels, inflammation, oxidative stress and rate of gluconeogenesis [63,83]. Another study using diabetic rats concluded that the supplementation of probiotic strain *L. plantarum* appreciably reduced serum α-amylase action, favoring the glycemic index mechanism by restricting carbohydrate absorption and hydrolysis [84]. The outcome of some of these animal studies is summarized in Table 1. 

### 3.2. Human Studies

In context to the elevated risk of T1D, early exposure to probiotic supplements might lead to a decrease in the risk of islet β-cell autoimmunity [8,85,86]. In addition, the use of probiotics in T1D adults has also been associated with a better glycemic control, increased synthesis of GLP-1 (beneficial insulinotropic gut hormone), and reduced TLR4 signaling (an inflammatory signaling) [87,88,89]. These changes in turn are associated with decreased incidences of T1D. In T1D children, supplementation with *L. rhamnosus* GG and *B. lactis* Bb12 at a dose of 10^9^ colony-forming units (CFUs) once daily for 6 months has been shown to regulate gut microbiota perturbations, thereby beneficially modulating the immune cells and preserving the number and proliferation of pancreatic β-cells [90]. It has also been reported that the administration of one capsule per day containing 10^8^ CFUs of LjN6.2 for 8 weeks in adult human subjects can regulate the infiltration of monocytes, natural killer cells, circulating Teff TH1 cells and cytotoxic CD8^+^ T cells in the islets; these changes may aid in the prevention of T1D occurrence [86]. Further, increased numbers of TH17 and TH1/TH17 cells have been found following probiotic treatment [86]. However, a significant increase in the concentration of IgA was also observed in the probiotics-treated vs. placebo group [86]. The use of probiotics-containing products by T1D adult patients (mean age 46 ± 14 years, 45% men) can help maintain better glycemic control and ameliorate conditions of metabolic syndrome, such as high blood pressure, high triglyceride levels, and lowered levels of high-density lipoprotein-cholesterol [91]. In addition, TEDDY studies from six clinical centers (three in the United states [Colorado, Georgia and Washington] and three in Europe [Finland, Germany, Sweden]) have found that early exposure of probiotic supplements (at the age of 0–27 days) could reduce the risk of T1D in children who had an increased risk of acquiring the disease [85]. Infant probiotic supplementation was provided either through dietary supplements or infant formula, which varies from nation to nation, but the median age for early exposure to probiotics was 42 days [85]. Most of Finnish children (n = 827 [95.2%]) received probiotics by dietary supplements whereas German infants (n = 241 [90.3%]) were exposed to probiotics through infant formula [85]. The outcome of some of these studies is summarized in Table 2. Altogether, these results suggest that early life exposure to probiotics can reduce the risk of T1D progression. However, not all probiotic strains show similar effects, as evidenced by another study performed on young children carrying the genetic risk of T1D, which found the intake of a specific probiotic strain during the first two years of life to be associated with an increased development of islet autoimmunity and progression of T1D [92]. Although the reason why such outcomes received from these studies remain unresolved, factors including the type and viable count of probiotic strain(s) used, host diet, as well as the stage of T1D autoimmunity progression at the time of intervention could be speculated on in order to understand how they influence these outcomes.

## 4. Prebiotic Interventions to Ameliorate T1D

Like probiotics, specific prebiotic substrates have also been reported in several human and/or animal studies to positively influence blood glucose levels by exerting hypoglycemic effects, thereby ameliorating T1D. Dietary prebiotics are selectively fermentable elements that are responsible for particular activity by bringing out precise changes in gut microbiota, thereby conferring specific health benefits to the host [93]. Presently, prebiotics are categorized into three major groups: fructooligosaccharides or galactooligosaccharides (GOS), lactulose, and non-digestible carbohydrates. The food ingredients considered non-digestible carbohydrates are large polysaccharides (inulin, resistant starches, cellulose, hemicellulose, pectins, and gums), some oligosaccharides that escape digestion, and unabsorbed sugars and alcohols [25]. Of these, the two most commonly used prebiotics are inulin-type fructans and GOS [25,94]. The most favorable and beneficial effect of prebiotics is promoting the growth and activity of specific beneficial gut microbes and maintaining (or restoring) healthy immunomodulation in the intestine [95]. Prebiotics can influence the gut microbiota in a way that could promote the development of healthy immune signaling in GALT and mucosal immune system [95] and alter the lymphoid immune expression by decreasing the production/level of pro-inflammatory cytokines while increasing that of anti-inflammatory cytokines [95]. This could also lead to increased expression of IL-10 and INF-α in PPs and secretory IgA, with a simultaneous increase in mucosal Ig [95]. Prebiotics can also provoke the proliferation of intestinal enteroendocrine L-cells to amend intestinal peptide secretions such as GLP-1 [96], which could help in modulating the intestinal inflammation of affected individuals [96]. In humans and animal models, the major fermented products of carbohydrate metabolism by intestinal microbiota are SCFAs, such as acetate (C2), propionate (C3) and butyrate (C4) [97]. These SCFAs (C2:C3:C4) are typically found in the molar ratio of 60:20:20 in the gut of both mice and humans [97]. Lactate is also produced in the gut but gets immediately converted into acetate [97]. It has been revealed that the *Bifidobacterium longum* strain produces acetate to facilitate protection against enteropathogenic infections [98]. In the lower intestine of humans [96] and rats [71,96], prebiotics could be particularly important for the production of SCFAs, which act as ligands for several GPCRs, such as FFAR2 and FFAR3. These receptors stimulate the enteroendocrine L-cells to trigger the release of GLP-1, which regulates gut permeability and helps in regulating inflammation [96] (Figure 4).

### 4.1. Animal Studies

Dietary fibers have been shown in several studies to be helpful in maintaining microbiota homeostasis and positively modulating gut permeability, thereby also delaying the development of T1D (Table 1). For example, the inoculation of long-chain inulin-type fructans [ITF](l), but not short-chain inulin-type fructans, in NOD mice has been shown to positively regulate gut-pancreatic immunity as well as gut barrier function and microbiota homeostasis, which results in delaying the progression of T1D [99]. ITF(l) intake was found to regulate cytokine production in the colon, pancreas and spleen [99], and also modulate gut microbiota with more species of *Ruminococcaceae* and Lactobacillus and an increased *Firmicutes* to *Bacteroidetes* ratio, to anti-diabetogenic levels [99]. In high-fat diet fed mice, dietary prebiotics supplemented with oligofructose (OFS) have been found to enhance *Bifidobacterium* counts [100]. Notably, increased intestinal levels of *Bifidobacterium* spp. have been found to be positively correlated with glucose-induced insulin-secretion, glucose-tolerance and decreased adipose tissue, pro-inflammatory cytokines and release of endotoxins [101] (Table 1). In alloxan-induced diabetic mice, caffeic-acid-rich fractions extracted from *Prunella vulgaris* L. could serve as a potential therapeutic agent to treat T1D [64]. Supplementation of human milk oligosaccharide (HMOS) in NOD-mice has also been found to prevent the occurrence of T1D by modulating the immune response via pancreas and MLNs [102]. Gut microbes including *Bifidobacterium (B.) infantis, B. bifidium, B. breve* and *B. longum* typically grow quite well in the presence of HMOS [33]. These bacteria interact with DCs, thereby modulating the systemic and mucosal immune systems by producing HMOS-derived SCFAs [102], which eventually leads to reduced levels of IL-17 and INF-γ with increased TNF-α levels during later stages of the disease [102]. TNF-α has been shown to reduce the destruction of pancreatic islet in NOD mice [102]. SCFAs can directly modulate pancreatic islet β-cell pathogenesis to reduce insulitis [56]. Moreover, in NOD-mice, acetate treatment has been found to result in a decreased occurrence of insulitis [69]. The availability of SCFAs, mainly butyrate, has been found to be positively associated with intestinal epithelial barrier function by producing more mucin [103], which has been known to increase intestinal barrier integrity and is plausibly partly responsible for protection against T1D [102]. Administration of dietary resistant starch to STZ-induced T1D rats has also been found to delay the progression of diabetic neuropathy by maintaining vitamin D balance [104]. Moreover, feeding a hydrolyzed casein-based diet to spontaneous mutation-induced diabetes-prone LEW.1AR1-*iddm* rats has been shown to modify the immune cell distribution and suppress the development of diabetes [105]. When NOD-mice are fed alternative dietary wheat sources, they show resistance to T1D because they lack T1D-linked epitopes. As a result, pro-inflammatory cytokines like INF-γ are decreased and the levels of anti-inflammatory cytokines (e.g., IL-10) are upregulated, which may hinder the development of T1D [106]. 

### 4.2. Human Studies

Table 2 summarizes some of the clinical studies of prebiotic interventions and their findings related to T1D-associated features and outcomes. In a pilot study on school-age children (aged 8–17 years), the consumption of prebiotics (oligofructose-enriched inulin) was found to positively influence gut microbiota, specifically increasing the number of *Bacteroidetes* and lactic acid-producing bacteria, which correlated with positive modulation of intestinal permeability and reduced inflammation, ultimately improving glycemic control and reducing the chances of T1D occurrence [109,113]. Supplementation of higher dietary fibers in adult T1D patients has also been found to decrease the chances of coronary heart disease by lowering systolic and diastolic blood pressure [110]. Furthermore, higher consumption of dietary fiber (>30 g/day) in T1D patients may also be associated with reduced inflammation [111]. Exposure to gluten during breastfeeding or infant stages in the form of gluten-containing cereals may also increase the risk of islet autoimmunity and T1D development [114]. In vitro studies have shown that the selective fermentation of prebiotics in response to the metabolism of fructan-like shorter-chain oligofructoses or longer-chain inulin compounds increases the abundance of four different clustered strains of *Bifidobacterium* [115]. These strains are associated with the production of acetate, which is responsible for protection from enteropathogenic infections. Acetate can alter the host immune system and ameliorate T1D [98] by leading to an increased production of anti-inflammatory cytokines and enhanced phagocytic effect by recovering the overall immune response in T1D individuals [98,116]. Breastfed infants exhibit decreased chances of developing autoimmune disease such as T1D [117]. The human milk provides long-chain polyunsaturated fatty acids (PUFA) to infants, which may act as a protective measure against T1D, owing to their association with decreased immune-antigens and primary islet autoimmunity [118]. On the other hand, the feeding of cow milk could increase islet autoimmunity among low/moderate risk HLA-DR genotypes, rather than high risk HLA-DR genotypes [118,119]. However, to date, there have not been any reports of T1D occurrence among calves, which might be because of the lack of research in this area and/or may be due to the BoLA-DQA1 gene, which has resistance to inflammation against microbial actions of *Streptococcus* and *Escherichia* species [120]. Therefore, human milk, most possibly because of its oligosaccharides and polysaccharides, could be responsible for reduced autoimmunity and amelioration of T1D.

## 5. Other Gut Microbiota Modulators in T1D

In addition to probiotics and prebiotics, several drugs have also been found to be helpful in maintaining blood glucose levels and intestinal cellular homeostasis. For instance, the supplementation of the antidiabetic drug metformin, in combination with probiotics and/or prebiotics, could modulate gut microbiota [107] and improve glycemic control with improved insulin sensitivity in C57Bl/6J mice [107]. Also, the inoculation of dapagliflozin (DAPA) in adult human subjects (12–21 years old) could help in controlling glycemic levels by reducing the required amount of inulin in the blood [112]. DAPA is a sodium-glucose co-transporter 2 (SGLT-2) inhibitor that acts by inhibiting SGLT-2 in the proximal tubule of kidneys. This decreases the reabsorption of glucose thereby increasing the secretion of glucose in urine [112], and has been found to increase the intestinal population of segmented filamentous bacteria, i.e., *Akkermansia muciniphila,* thereby providing protection against autoimmune T1D disease, as mentioned in Table 1 [121]. The drugs Artemether and GABA act through GABA receptors and have been associated with the proliferation of β-cells, thus reducing T1D incidence in mice [122]. The treatment of T1D human islet cells with GABA was found to be responsible for the proliferation of β-cells by interfering with the NF-κB signaling pathway [108]. Ketones, such as acetoacetate and DL-β-hydroxybutyrate, have also been deemed effective in activating NADPH oxidase, which increases oxidative stress and reduces cellular injury, preventing a higher risk of T1D complications [123].

## 6. Conclusions and Future Perspectives

The risk of T1D at birth can be due to abnormal immunomodulation impacted by several environmental factors, such as gut microbiota or exposure to different diets and infections, which ultimately leads to the development of insulitis with severe T1D. Macrophages and cytotoxic T lymphocytes (particularly CD8^+^ cells) are responsible for pancreatic β-cell damage and further adverse effects created by cytotoxic CD20^+^ cells. Moreover, an irregular proliferation of innate natural killer cells and interleukins are associated with autoimmune T1D by provoking an inappropriate adaptive immune response. The defective functioning of the adaptive immune response can be developed by hampering the normal activity of innate immunity in GALT due to the presence of a hostile gut microbiota. Therefore, the development of therapies to modulate gut microbiota and innate immune functions in the GALT could be one of the ideal strategies to prevent T1D pathogenesis in its initial stages. Specific probiotic microbes such as *Lactobacillus*, *Bifidobacterium*, *Saccharomyces boulardii* and other beneficial groups of microbes as well as prebiotics such as inulin, resistant starches, cellulose, pectins, and others have been found to be beneficial against this autoimmune condition. Most mechanisms of these agents are associated with preventing the growth of hostile gut microbiota and stimulating intestinal homeostasis, which is characterized prominently by reduced intestinal inflammation. Therefore, future studies should focus on disentangling the gut microbial clades associated with T1D development, which, although challenging, could prove to be helpful in understanding the causes of T1D. Undoubtedly, this particular autoimmune disease requires well-coordinated all-inclusive investigations to determine the role of intestinal microbiota in its development. Indeed, well-characterized and high-quality longitudinal sampling and data collection starting from pregnancy to the onset of T1D in high risk children might facilitate in tracking the mechanisms associated with the microbiota’s effect on immune response and altered gut integrity (leaky gut), which are common determinants of T1D. Moreover, further research is necessary to investigate the role and functionality of gut microbiota in particular context to the prevention and protection against T1D by analyzing the functional pathways of the microbial and host transcriptomes, metabolomes and proteomes involved in T1D development. Establishing and validating functional regimens, like probiotics and prebiotics, in long-term clinical studies is necessary. Indeed, studies focused on the pathways of the host immune response that could provide novel therapeutic approaches for early diagnosis of T1D, while also supplying knowledge for curing this autoimmune disease, are warranted.

## Figures and Tables

**Figure 1 microorganisms-07-00067-f001:**
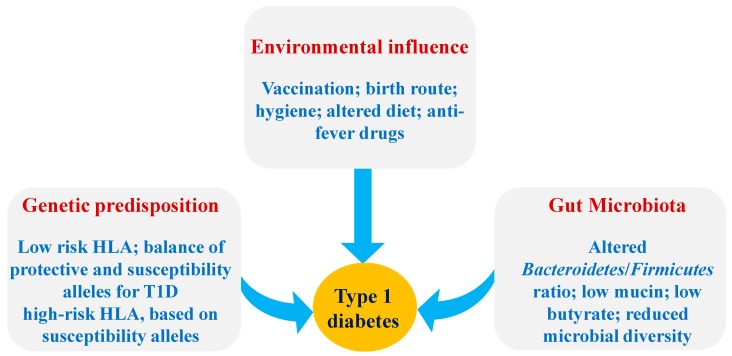
Factors influencing the susceptibility of T1D. Abbreviations: HLA: Human leukocyte antigen; T1D: Type 1-Diabetes.

**Figure 2 microorganisms-07-00067-f002:**
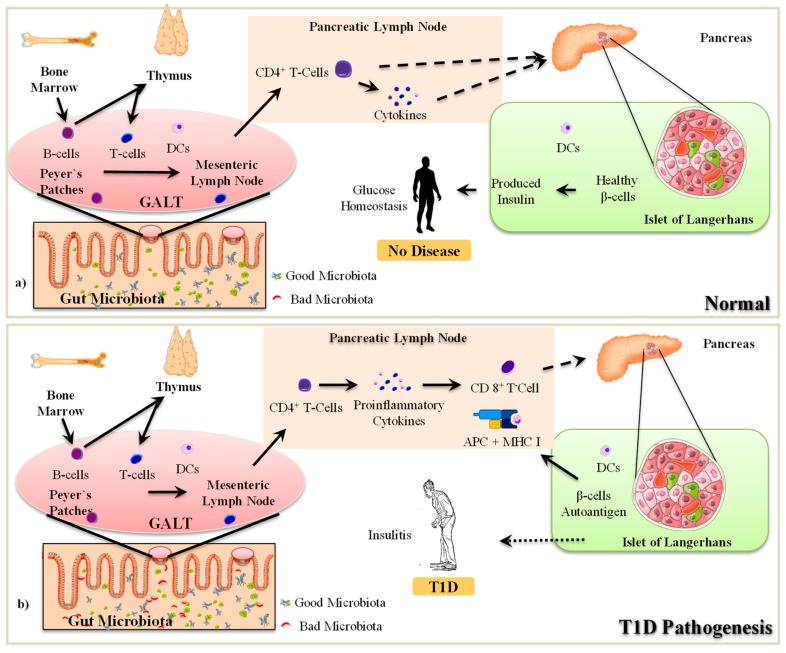
Mechanisms involved in the pathogenesis of T1D. APC: Antigen presenting cell; DCs: Dendritic cells; GALT: Gut Associated Lymphoid Tissue; MHC: Major Histocompatibility Complex; CD 8^+^ T-Cell: Cytotoxic T lymphocytes; CD4^+^ T-Cells: Helper T lymphocytes; T1D: Type-1-Diabetes.

**Figure 3 microorganisms-07-00067-f003:**
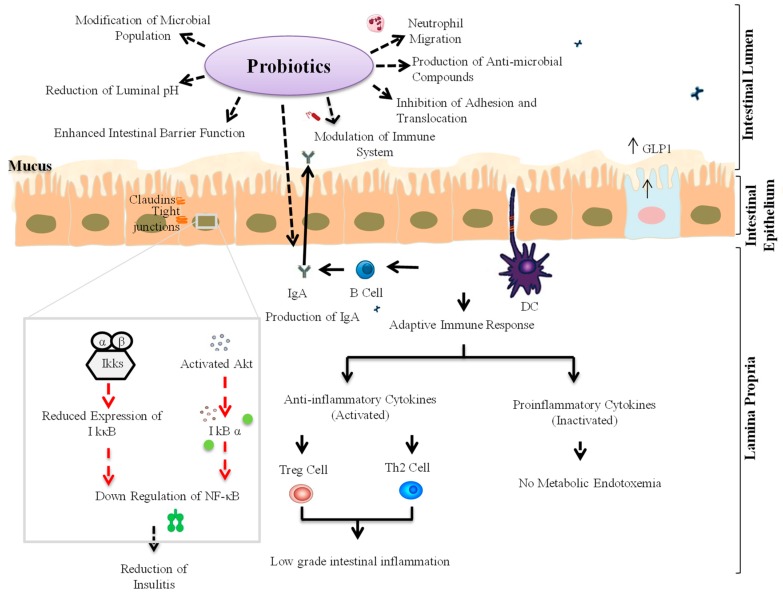
Schematic representations of mechanisms of actions through which specific probiotic strains might help in the amelioration of T1D. Akt: protein kinase B; DCs: Dendritic cells; GLP-1: Glucagon-like peptide-1; NF-κB: Nuclear factor kappa-light-chain-enhancer of activated B-cells; IκBα: I kappa B kinase; IkkB: IκB kinase beta; IgA: Immunoglobulin A; Treg: T regulatory cell; Th2: T-helper cell 2.

**Figure 4 microorganisms-07-00067-f004:**
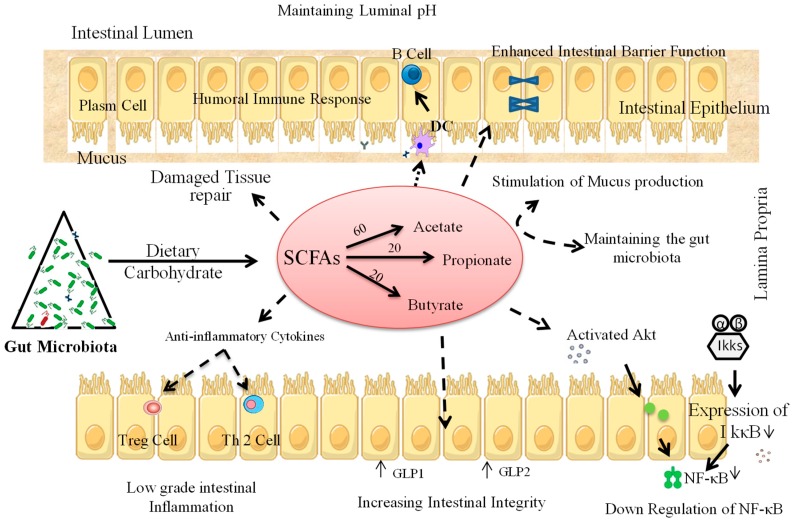
Purported mechanism(s) of action through which prebiotics could manipulate gut microbiota as well as immune cells of T1D pathology. Akt: protein kinase B; DC: dendritic cell; GLP-1: glucagon-like peptide-1; GLP-2: glucagon-like peptide-2; IkkB: IκB kinase beta; IκBα: I kappa B kinase; NF-κB: Nuclear factor kappa-light-chain-enhancer of activated B-cells; SCFAs: short chain fatty acids; Th2: T-helper cell 2; Treg: T regulatory cell.

**Table 1 microorganisms-07-00067-t001:** Summary of major animal studies of probiotic and prebiotic interventions and their findings related to T1D-associated features and outcomes.

Probiotics/Prebiotics	Model Type	Mechanism of Action	Major Findings	References
Oral Probiotics VSL#3 (*Bifidobacteriaceae*, *Lactobacillaceae*, *Streptococcus thermophilus*)	NOD mice	- Generates more pro-tolerogenic components of inflammasome like indoleamine 2,3-dioxygenase (IDO) and IL-33. - Reduces the synthesis of inflammatory cytokines like IL-1β. - Promotes CD103^+^ differentiation. - Reduces Teff/Treg cell ratios within the gut mucosa, MLNs and PLNs.	- Modification of gut microbial environment. - Modulating T1D pathogenesis.	[73]
Bacterial LPS or Zymosan	NOD mice	- Produces synergetic innate immune response through TLR2 and Dectin-1 signaling. - Eliminates inflammatory immune cells and suppresses autoimmunity. - Triggers the secretion of immune regulatory factors like IL-10, TGF-β1, IL-2 and Raldh1A2. - Increases the numbers of Foxp3^+^CD4^+^ T cells in the PLN but not in spleen.	- Used as an immune regulatory adjuvant for promoting β-cell antigen-specific immune modulation. - Reverses the early stages of hyperglycemia in T1D.	[47]
*Lactobacillus brevis* KLDS 1.0727 and KLDS 1.0373	STZ-induced C57BL/6 T1D mice	- High GABA generating capacity due to the gad gene. - Significant effect in lowering the blood glucose level or insulin in plasma.	- Inhibits the development of T1D in diabetic mice model.	[62]
PFM with 1% of *Lactobacillus* speci*es*	STZ-induced albino wistar T1D rats	- Significant decrease in the expression of hepatic gluconeogenesis gene like Glucose-6-phosphatase (G6Pase) and Phosphoenol pyruvate carboxykinase (PEPCK) in the liver. - Significant reduction in serum inflammatory cytokines like IL-6 and TNF-α -Decrease in HbA1_c_, blood glucose level and serum lipid profile. -Significant increase in the serum insulin level.	- Increases insulin level with significant reduction in blood glucose level. - Improvement in glucose metabolism -Decrease in inflammation, oxidative stress and hepatic gluconeogenesis.	[83]
HMOS Prebiotic	NOD Mice	- Increases SCFA concentration in the gut. - Limits autoimmune T-cells and increases the Treg cells. - Induces tolerogenic DC phonotype by induction of MHC II and increases the expression of inhibitory molecules such as PD-L1 and OX40-L. -Increased butyrate production promoting mucin synthesis. - Improves intestinal barrier integrity. -Reduced pancreatic islet destruction by regulating the immune system.	- Modulation and maintaining the α- and β-diversity of the fecal microbiota. - Changes the direct shape of the pancreatic environment, resulting in less insulitis. - Helps in protection against T1D.	[102]
Dietary Resistant starch	STZ-induced T1D Sprague-Dawley rats	- Influences the secretion of GLP-1 and PYY hormones. - Proliferation of β-cell and insulin synthesis. - Provides nephron-protection. - No effect on blood glucose level and Vitamin D balance.	- Develops normalized growth pattern in T1D.	[104]
CARF extracted from PV	Alloxan-induced T1D Swiss Webster mice	-Decreases α-amylase and α-glucosidase activity. - Reduces HbA1c level. - Elevates serum insulin level. - Increases antioxidant enzyme level.	- Poses anti-diabetogenic, anti-nociceptive and hypoanalgesic properties as therapeutic agents against T1D.	[64]
Prebiotic oligofructose	High fat diet induced diabetic C57b16/J mice	-Increases *Bifidobacteria* number by modifying gut microbiota. - Decreases endotoxemia. -Improves glucose tolerance and regulates glucose-induced insulin secretion. -Increases colonic GLP-1 secretion.	-Pathophysiological regulation of endotoxemia. - Sets the tone of inflammation, glucose tolerance and insulin secretion.	[101]
Oral transfer prebiotic *Lactobacillus johnsonii* N6.2-Mediated	KRV virus induced-BBDP rat	- TH17 lymphocyte biasness within the gut-draining MLN. - Cytokines, like IL-6 and IL-23, were responsible for induction and sustenance of TH17 cells was higher. - Retention of TH17 differentiation state that may prevent T-cell conversion to the diabetogenic phenotype.	- Confirms resistance to T1D.	[65]
Probiotic *Bifidobacterium* spp.	STZ-induced C57BL/6J diabetic mice	- Significant reduction in blood glucose level. - Increases the protein expression of insulin receptor β, insulin receptor substrate 1, (Akt/PKB), IKKα, IκBα. - Decreases the macrophage chemoattractant protein-1 and IL-6 expression.	- Responsible for treating diabetes.	[72]
*Lactobacillus reuteri*	STZ-induced C57BL/6 diabetic mice	- Development of anti-inflammatory property by inhibiting osteoblast TNF-α signaling. - TNF-α modulates the Wnt10b expression in T1D.	- Use of probiotic to benefit bones in T1D patients.	[74]
*Lactococcus lactis*	NOD Mice	- Increases the frequency of local Tregs in the pancreatic islet. - Suppresses immune response in an autoantigen-specific way. - Preserves functional β-cell mass and reduces insulitis. - Secretion of human pro-insulin and IL-10 can stably revert autoimmune diabetes. - Induced Ag-specific Foxp3^+^ Tregs that prevent diabetes transfer.	- Treatment strategy for T1D in humans.	[76]
*Lactobacillus kefiranofaciens and Lactobacillus kefiri*	STZ-induced C57BL/6 diabetic mice	- Level of IL-10 significantly raised in pancreas. - Increased IL-10 inhibits the secretion of pro-inflammatory cytokines, like TNF-α and TH1 (also IL-1β, IL-2, IL-6).	- Potential ability to stimulate the release of GLP-1.	[78]
Bifibobacteria, lactobacilli *and Streptococcus salivarius subs.*	NOD mice	- Decreases the rate of β cell destruction. - Increases the production of IL-10 from PPs, pancreas and spleen. - Modulates GALT.	-Prevention of autoimmune diabetes. -Induces immunomodulation by a reduction in insulitis severity.	[79]
*Lactobacillus johnsonii* N6.2	T1D BBDP rats	- Changes in the native gut microbiota. - Induced changes in host mucosal protein and oxidative stress response. - Decreases oxidative response protein in the intestinal mucosa. - Decreases pro-inflammatory cytokines, like IFN-γ. - Higher expression of tight junction proteins, like claudin.	- Delays or inhibits the occurrence of T1D.	[82]
*Lactobacillus plantarum* TN627	Alloxan induced-diabetic rat	- Improved the immunological parameters of the pancreas. - Reduced the pancreatic and plasmatic α-amylase activity as well as blood glucose level. - Decreased the pancreatic and plasmatic lipase activity, serum triglyceride and LDL-cholesterol rate. - Increases the HDL-cholesterol rate.	- Helpful in preventing diabetic complications in the adult rat.	[84]
Low antigen, hydrolyzed casein-based diet	LEW.1AR1-*iddm* Rat model	- Increased immunoregulatory capacity and gut immune deficits. - Decreased expression of CD3^+^ T-cells, CD163^+^ M2 macrophages and Foxp3^+^ cells in jejunum. - Decrease in CD4^+^ Foxp3^+^ regulatory T-cells in PLNs. - IFN-γ expression increase in MLNs.	- Protection against T1D.	[105]
*Bifidobacterium animalis* ssp. *lactis* 420 (B420) and Metformin	Ketogenic diet-induced C57Bl/6J diabetic mice	- Increases ileum GLP-1 concentration. - Increases the amount of insulin released from pancreatic β-cells. - Significantly decreases the glycemic response and plasma glucose concentration.	- Improves glucose metabolism and insulin secretion. - Improves the efficacy of metformin.	[107]
Wheat Flour	NOD mice	- Lacks the epitopes linked with T1D. -Reduction in the level of pro-inflammatory cytokines, like IFN-γ. -Increase in the level of anti-inflammatory cytokine IL-10.	- Reducing the incidence of T1D.	[106]
Systemic GABA therapy	STZ-induced C57/BL6 T1D mice	- Increases klotho (anti-aging agent) level expression in serum, pancreatic Islet of Langerhans and kidneys. - Klotho stimulates pancreatic β-cells survival and proliferation. - Increases insulin secretion. - Klotho blocks NF-κB activation by interfering with its nuclear translocation. - Suppresses autoimmune responses.	- Important implications for the treatment of T1D.	[108]
Dietary fibers	NOD mice	- Increases CD25^+^Foxp3^+^CD4^+^ Treg and decreases IL17A^+^CD4^+^Th17 cells. - Changes the cytokine production profile in the pancreas, spleen and colon. - Enhances tight junction proteins (claudin-2, occludin) and SCFAs. - Enhances *Firmicutes/Bacteroidetes* ratio as well as *Ruminococcaceae* and *Lactobacilli.*	- Modulates T-cell response. - Modulates gut-pancreatic immunity. - Delays the development of T1D.	[99]

Abbreviations: Ag: antigen; Akt/PKB: protein kinase B; BBDP: bio-breeding diabetic pathogen; DC: dendritic cell; GABA: gamma-aminobutyric acid; GALT: gut-associated lymphatic tissue; GLP-1: glucagon-like peptide-1; HDL-cholesterol: high-density lipoprotein-cholesterol; HbA1c: hemoglobin A1c; HMOS: human milk oligosaccharide; IκKα: IκB kinase alpha; IκBα: Nuclear factor-kappa B inhibitor alpha; IL: interleukin; KRV: Kilham rat virus; LDL-cholesterol: low-density lipoprotein-cholesterol; LPS: lipopolysaccharide; MHC: major histocompatibility complex; MLN: mesenteric lymph node; NOD: non-obese diabetic; PLN: pancreatic lymph node; PP: Peyer’s patches; PYY: peptide YY; SCFA: short-chain fatty acid; STZ: Streptozotocin; T1D: type-1 diabetes; Teff: effector T-cell; TGF: transforming growth factor; Th17 cell: T-helper cell 17; TLR: toll-like receptor; TNF: tumor necrosis factor; Treg: regulatory T-cell.

**Table 2 microorganisms-07-00067-t002:** Summary of major human studies of probiotic and prebiotic interventions and their findings related to T1D-associated features and outcomes.

Probiotics/Prebiotics	Model Type	Mechanism of Action	Major Findings	References
*Lactobacillus johnsonii* N6.2	42 healthy adult humans	- Increased serum tryptophan level -Resulted in a decreased kynurenine:tryptophan ratio. - After washout period, monocytes and natural killer cell numbers increase significantly regulated through indoleamine 2,3-dioxygenase (IDO) pathway. - Increases circulatory effector Th1 cells and cytotoxic CD8^+^ T-cells. - Delay or reduces the apoptosis of memory CD8^+^ T-cells.	- Responsible for reducing the risk of T1D occurrence.	[86]
*Lactobacillus rhamnosus GG and Bifidobacterium lactis Bb12*	96 children aged between 8–17	- Improved the gut mucosal barrier. - Modulated local and systemic immune responses. - Reduced the risk of autoimmunity.	-Inhibits the growth of pathogens -Preserves the β-cell function.	[90]
*Probiotics*	1039 adult individuals	- Decrease in obesity, body mass index, waist-to-hip ratio. - Regulated blood pressure, HDL-cholesterol, triglyceride components. - Significantly associated with better glycemic control.	- Beneficial effect on various factor related to the diabetic complications.	[91]
Prebiotic (Oligofructose-enriched inulin)	Young children aged between (8–17 years)	-Develops into severe hypoglycemia. -Decreases endotoxemia and reduced insulin resistance. -Improves glycemic control. - Changes gut microbiota, permeability and inflammation.	- A potential and novel agent for treating T1D.	[109]
Dietary fiber intake	T1D adult human patients	- Exhibited lower systolic and diastolic blood pressure. - No significant association was found in lipid profile. - Shows lower Body Mass Index (BMI), superior metabolic control of diabetes. - Reduction in the use of medicine to treat diabetes (insulin) and hypertension (ACE inhibitor).	- Association with lower blood pressure in T1D patients.	[110]
Dietary fiber	106 outpatients with T1D	- Develops anti-inflammatory properties. -Decreases the C-reactive protein levels independent of HbA1c value.	- Plays a significant role in reduction of inflammation. - Associated with lowering the risk of coronary heart disease.	[111]
Adjunct therapy with DAPA	33 Youth T1D patients aged between 12–21 years	- Reduced the mean insulin requirement dose for medication. - Increase in urinary glucose excretion. -Leads to a significant reduction in insulin requirements to achieve the target glucose level, irrespective of HbA1c level.	- Offers a future therapeutic agent to the T1D challenged pediatric age group.	[112]

Abbreviations: ACE: angiotensin-converting-enzyme; DAPA: dapagliflozin; HbA1c: hemoglobin A1c; HDL-cholesterol: high-density lipoprotein-cholesterol; T1D: type-1 diabetes; Th1 cell: T-helper cell 1.

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
