# Peer review of "Probiotics and Prebiotics for the Amelioration of Type 1 Diabetes: Present and Future Perspectives"

_microorganisms, 2019, doi:10.3390/microorganisms7030067_

Round 1

Reviewer 1 Report

In this review, Mishra et.al.  have presented a comprehensive perspective on the role of microbial perturbation on T1D and its restoration as a potential remedy for the disease. In general, the review is well written and covers a lot of ground. I have a few suggestions and conceptual questions to improve the narrative and making the manuscript more attractive for the readers:

Figure 1: Authors mention Bacteroidetes/Firmicutes ratio in the figure. While this reviewer still feels that this ratio can be utilized as an indicator of dysbiosis in specific contexts, the concept itself has fallen into some controversy. I think the authors should address this controversy in their manuscript.

It might be pertinent to draw a discussion between other inflammatory complications in the pancreas (e.g. pancreatic cancer), which have some immunogenic similarities with T1D, where microbiome seems to play a major role. Any insight on B7-H4 and microbiome? [PMIDs: 29567829, 24326367].

Lines 263-283: The role of bacterial metabolites is gaining immense attention in the context of several diseases. More so in metabolic disorders. I would suggest the inclusion of PMIDs 30039798, 28401628, 28069754 and 28061847, where the authors would appreciate that many more bacterial metabolites could potentially affect T1D in progression as well as amelioration. 

The use of probiotics mentioned in this section (lines288-345): Are they of similar composition? If not, how does one reconcile with the outcomes? 

Lines 374-375: The word 'specific' appears way too many times- please consider revision.

Line 444: please elaborate 'positively influence the gut microbiome'. Increase diversity?

The arguments presented within lines 459 and 463 are based on highly correlative data and needs to be better resolved. If this simple logic is to be followed, some mention should be made as to why calves do not have T1D.

Well articulated discussion.

Author Response

REVIEWER #1

In this review, Mishra et.al.  have presented a comprehensive perspective on the role of microbial perturbation on T1D and its restoration as a potential remedy for the disease. In general, the review is well written and covers a lot of ground. I have a few suggestions and conceptual questions to improve the narrative and making the manuscript more attractive for the readers:

AUTHORS: Thank you very much for your encouraging feedback and time to review our paper. We have tried our best to include all your suggestions in the revised manuscript and point-by-point response is addressed below.

Figure 1: Authors mention Bacteroidetes/Firmicutes ratio in the figure. While this reviewer still feels that this ratio can be utilized as an indicator of dysbiosis in specific contexts, the concept itself has fallen into some controversy. I think the authors should address this controversy in their manuscript.

AUTHORS: Interesting point, we should discuss about this, with figure 1.

It might be pertinent to draw a discussion between other inflammatory complications in the pancreas (e.g. pancreatic cancer), which have some immunogenic similarities with T1D, where microbiome seems to play a major role. Any insight on B7-H4 and microbiome? [PMIDs: 29567829, 24326367].

AUTHORS: Thanks for nice point. We have addressed this in revised manuscript.

Lines 263-283: The role of bacterial metabolites is gaining immense attention in the context of several diseases. More so in metabolic disorders. I would suggest the inclusion of PMIDs 30039798, 28401628, 28069754 and 28061847, where the authors would appreciate that many more bacterial metabolites could potentially affect T1D in progression as well as amelioration.

AUTHORS:  Yes, it’s a good point, and we have included in the revised manuscript.

The use of probiotics mentioned in this section (lines288-345): Are they of similar composition? If not, how does one reconcile with the outcomes?

AUTHORS: We have included the available information from literature. The composition of probiotics are not similar, which is generally expected, however, we included the best available information.

Lines 374-375: The word 'specific' appears way too many times- please consider revision.

AUTHORS: Sorry, we have revised this.

Line 444: please elaborate 'positively influence the gut microbiome'. Increase diversity?

AUTHORS: We have revised this part to make it more clear.

The arguments presented within lines 459 and 463 are based on highly correlative data and needs to be better resolved. If this simple logic is to be followed, some mention should be made as to why calves do not have T1D.

AUTHORS: Your point is well taken and explain in appropriate way in revised manuscript.

Well-articulated discussion.

AUTHORS: Thanks a lot for your compliments.

Reviewer 2 Report

This article reviews the potential interactions between the gut microbiome and immune mechanisms that are involved in the progression of T1D. The review offers to understand the potential effects and prospects of gut microbiome modulators including probiotics and prebiotics interventions in the amelioration of T1D pathology in both humans and animal models.

This is a well-written article and provides valuable information for those working in this area. It would be more interesting if the review includes the interconnection roles between prebiotics and probiotics and whether any study combined both the functional regimens.

Minor comments are given as follows:

1)      Please review the objective in the abstract and in the text to ensure consistency

2)      Introduction

a.       Suggest adding information about any other similar review of its kinds and how this current review extend reader understanding of the topic

b.       The significance of the review to the reader

3)      A methodology section is encouraged for a narrative review to enable the reader to consider selection bias in manuscript selection. The process is generally not as rigorous as that for a systematic review, but years searched, databases searched, and publication bias should be mentioned.

4)      Figure 2: Proposed or confirmed mechanism? From the author’s work or summary from various work?

5)      Line 252-256: A well-discussed section

6)      Line 360-369: How about study duration, dosage and whether required a dose-response prescription?

7)      As it is clear that there is a difference between human and animal model as elaborated in line 255, how knowledge and understanding obtained from animal model contribute to human trials

8)      Table 1- The content arrangement is relatively difficult to understand. Suggest dividing the table to animal and human studies based on the argument in no. 7. In the table, suggest standardizing the use of verb or noun to describe Major Finding or Mechanism of action.  

Author Response

REVIEWER #2

This article reviews the potential interactions between the gut microbiome and immune mechanisms that are involved in the progression of T1D. The review offers to understand the potential effects and prospects of gut microbiome modulators including probiotics and prebiotics interventions in the amelioration of T1D pathology in both humans and animal models.

This is a well-written article and provides valuable information for those working in this area. It would be more interesting if the review includes the interconnection roles between prebiotics and probiotics and whether any study combined both the functional regimens.

Minor comments are given as follows:

1)      Please review the objective in the abstract and in the text to ensure consistency

AUTHORS: Thanks for bringing this point up and we have revised manuscript to keep it consistant.

2)      Introduction

a.       Suggest adding information about any other similar review of its kinds and how this current review extend reader understanding of the topic

b.       The significance of the review to the reader

AUTHORS: These points have been addressed and new references are included.

3)      A methodology section is encouraged for a narrative review to enable the reader to consider selection bias in manuscript selection. The process is generally not as rigorous as that for a systematic review, but years searched, databases searched, and publication bias should be mentioned.

AUTHORS: Thanks, this information has been included in revised manuscript.

4)      Figure 2: Proposed or confirmed mechanism? From the author’s work or summary from various work?

AUTHORS: We have compiled all the figures as original without modifications or adoption from any prior published literature. All these mechanisms are drawn on the basis of multiple publications, and citing one or two article might not be appropriate to direct the readers.

5)      Line 252-256: A well-discussed section

AUTHORS: Thanks for your nice complements

6)      Line 360-369: How about study duration, dosage and whether required a dose-response prescription?

AUTHORS: This information has been included in revised manuscript.

7)      As it is clear that there is a difference between human and animal model as elaborated in line 255, how knowledge and understanding obtained from animal model contribute to human trials

AUTHORS: Humans and animals show discrepancies in some aspects of gut microbiome and immunity, however we tried to put most of the reports/information in this article that are common in humans and animals. However, mechanistic studies are more clear in animal models than humans, hence to better explain mechanisms, animal studies are more relevant than humans.

8)      Table 1- The content arrangement is relatively difficult to understand. Suggest dividing the table to animal and human studies based on the argument in no. 7. In the table, suggest standardizing the use of verb or noun to describe Major Finding or Mechanism of action. 

AUTHORS: Thank you it was a good idea. We divided this into two tables: Table 1: Mice studies; Table 2: Human studies.